

# Quercetin reduces hydroxyurea induced cytotoxicity in immortalized mouse aortic endothelial cells

Zachary M. Kiser[1], Monica D.M. McGee[2], Racquel J. Wright[3], Alexander Quarshie[4], Gale W. Newman[1], Karen R. Randall[5], Jonathan K. Stiles[1], Adel Driss[6] and Jacqueline M. Hibbert[1]

[1] Department of Microbiology, Biochemistry, and Immunology, Morehouse School of Medicine, Atlanta, GA, United States of America
[2] Spelman College, Atlanta, GA, United States of America
[3] Biotechnology Centre, University of the West Indies, Mona, Jamaica
[4] Community Health & Preventive Medicine, Morehouse School of Medicine, Atlanta, GA, United States of America
[5] Department of Pharmacology & Toxicology, Morehouse School of Medicine, Atlanta, GA, United States of America
[6] Department of Physiology, Morehouse School of Medicine, Atlanta, GA, United States of America

Corresponding authors
Zachary M. Kiser, zkiser@msm.edu
Jacqueline M. Hibbert, jhibbert@msm.edu

## ABSTRACT

**Background**. Chronic inflammation is a characteristic of sickle cell disease (SCD), and is invariably associated with vascular endothelial injury. Hydroxyurea (HU), a naturally cytotoxic chemotherapeutic agent, is the only FDA drug approved for SCD, and is therefore naturally cytotoxic. Quercetin (QCT) is a dietary flavonoid found ubiquitously in plants and foods that have anti-oxidative and anti-inflammatory characteristics. Our hypothesis is that dietary QCT will decrease cytotoxic effects of lipopolysaccharide (LPS) and HU induced vascular cell damage.

**Methods**. Lipopolysaccharide (LPS) was used to induce inflammation in immortalized mouse aortic endothelial cells (iMAECs), providing an in vitro model of inflamed endothelial cells. The cells were exposed to LPS throughout the entire experiment. Interventions included treating the LPS exposed cells with QCT, HU, or QCT + HU over 50 hours. The 50-hour period included 24 hours of varying treatments, followed by two hours of hypoxic exposure and then 24 hours under normal aerobic exposure.

**Results**. LDH level was significantly higher for LPS treated versus untreated cells ($P = 0.0004$). LPS plus 30 micromole QCT reduced the LDH ($p = 0.1$, trend), whereas LPS plus 100 micromoles HU, significantly increased LDH ($p = 0.0004$). However, LPS plus treatment with 30 micromoles QCT/100 micromoles HU, significantly reduced LDH, compared with HU alone ($p = 0.0002$).

**Discussion**. These results suggest that quercetin may be effective against vascular endothelial cell damage for iMAECs *in vitro*. In particular, it shows promise in preventing HU-induced cytotoxicity, surprisingly found from these results. This latter finding is important, and should be given more consideration, since HU is the only FDA-approved drug for treating sickle cell patients, and its use is rapidly increasing.

## INTRODUCTION

Sickle Cell Disease (SCD) is a genetic disease caused by a single nucleotide mutation in the beta globin gene, which increases the tendency of hemoglobin to polymerize and causes red blood cells (RBC) to acquire a sickle shape (*Pauling & Itano, 1949*). Accumulation of the polymerized hemoglobin S (HbS) within the RBC causes injury to the red cell, altering both its form and function. Accumulation of the damaged red blood cells within the blood vessels causes decreased blood flow and hence, low oxygen delivery. The low oxygen levels promote more red cell deformation, and increased polymerization in the already damaged red cells (*Steinberg, 2008*). As these damaged cells traverse the circulatory system they encounter and injure vascular endothelial cells. Hydroxyurea (HU) is the only FDA approved drug for SCD. It is a chemotherapy drug approved for use at low dose in SCD patients to increase the production of fetal Hb, which does not polymerize, therefore maintaining normal red cell function and life span and reducing the damage caused by the sickle cells. It is important to note that the cytotoxic nature of the drug, may present a problem for long term use as an SCD treatment (*Brun et al., 2003*; *Baz et al., 2012*).

Vascular endothelial cells serve multiple physiological functions in normal individuals. These include control of vascular tone, separating blood from the interstitial spaces and the presence of molecules and receptors involved in vascular adhesion, in both health and disease (*Hebbel, Osarogiagbon & Kaul, 2004*; *Rajendran et al., 2013*). Chronic inflammation in sickle cell patients is invariably associated with injury to the vascular endothelium. In SCD this damage leads to continuously varying activated states of vascular endothelial cells. System-wide inflammation accompanies endothelial cell activation. Increased adhesion of leukocytes to the vascular endothelium demonstrates evidence of this inflammatory state and is associated with shortening of the white cell's life span, and acquisition of a pro-adhesive phenotype by the vascular endothelial cells. Oxidative stress and reactive oxygen species also develop in the vascular endothelium (*Hebbel, Osarogiagbon & Kaul, 2004*; *Rajendran et al., 2013*). The effects of activation and inflammation lead to further complications within the endothelial cells, which include increased levels of inflammatory cytokines and adhesion molecules, hypoxia-reperfusion injury, increased reactive oxygen production and possible vaso-occlusion (*Osarogiagbon et al., 2000*; *Koo et al., 2001*; *Hebbel, Osarogiagbon & Kaul, 2004*; *Cook-Mills, Marchese & Abdala-Valencia, 2011*; *Chirico & Pialoux, 2012*).

Ischemia/reperfusion injury (IRI) occurs when the blood supply returns to tissue that was previously subjected to ischemia. Reflux of oxygen and nutrients to the formerly ischemic tissue destroys cellular DNA, specific proteins and the plasma membrane. The damage related to IRI stems from oxidative stress, which causes a non-specific inflammatory state in the vasculature. Ischemia/reperfusion injury also induces the production of oxygen free radicals. The increased production of oxygen free radicals causes the formation of reactive oxygen species. The rapid buildup of free radicals during the reperfusion phase of IRI causes the body's natural sources of antioxidants, such as superoxide dismutase and glutathione, to be depleted during the period of ischemia (*Koo et al., 2001*). The depletion of natural antioxidants allows oxidative damage to progress unchecked causing further injury to

the endothelium. A supplemental antioxidant supply, such as may be obtained from the flavonoid quercetin (QCT), may help to prevent the increasing free radical production (*Hyacinth, Gee & Hibbert, 2010*) as well as the damage to macromolecules, including DNA (*Potenza et al., 2008*).

The literature describes quercetin as a reliable anti-oxidant for *in vitro* studies (*Potenza et al., 2008*; *Boots, Haenen & Bast, 2008*; *Dajas et al., 2015*; *D'Andrea, 2015*; *Sharma et al., 2015*). The enzyme Xanthine Oxidase is one source of reactive oxygen species production (*Kelley et al., 2010*; *Cantu-Medellin & Kelley, 2013*) and reports in the literature show that QCT is an effective inhibitor of xanthine oxidase (*Middleton, Kandaswami & Theoharides, 2000*; *Pauff & Hille, 2009*; *Cao, Pauff & Hille, 2014*; *Bindoli, Valente & Cavallini, 1985*). The use of QCT as a supplemental source of anti-oxidants may help to reduce the drain on the already weak anti-oxidant levels in SCD patients. Based on the antioxidant properties of QCT, in the present study we investigated whether QCT may decrease the level of cytotoxicity associated with HU treatment on an inflammation/hypoxia model.

## MATERIALS AND METHODS

### Culture of immortalized mouse aortic endothelial cells

Immortalized Mouse Aortic Endothelial Cells (iMAEC), a stable cell line derived from endothelium of the mouse aorta, were obtained through a generous gift from the Jo Lab at Emory University (Atlanta, GA) (*Ni et al., 2014*). Cells were grown in Dulbecco's Modified Eagle Medium (DMEM) containing 10% fetal bovine serum, 1% 100X MEM Non-Essential Amino Acids Solution (MNEAA; Gibco, Waltham, MA, USA), 1.3% Endothelial Cell Growth Supplement (ECGS) (final concentration 50 μg/ml; Gibco, Waltham, MA, USA) and 1% 100X penicillin-streptomycin, following previously published procedures (*Koo et al., 2001*). The cultures were kept in a humidified chamber at 5% $CO_2$ and 37 °C, and the medium was changed every 1–2 days.

### Treatment of immortalized mouse aortic endothelial cells

iMAEC were seeded at a density of 10,000 cells/well and allowed to grow to approximately 70% confluency, which occurred in approximately 36 h. After the desired confluence was achieved, the iMAEC were treated with 250 ng/ml of lipopolysaccharide (Sigma-Aldrich, St. Louis, MO, USA) along with varying concentrations of Quercetin (M.W. 302.24 g/mol) (QCT) (Sigma-Aldrich, St. Louis, MO, USA), Hydroxyurea (M.W. 76.0547 g/mol) (HU) (Sigma-Aldrich, St. Louis, MO, USA) or a combination of Quercetin + Hydroxyurea (QCT+HU). The treatment with QCT, HU, or QCT+HU, lasted for 24 h. The cells were then subjected to two hours of enzymatic hypoxia, which is a model for ischemia-reperfusion injury, achieved using a combination glucose oxidase (GOX) (Sigma-Aldrich, St. Louis, MO, USA) and catalase (CAT) (Sigma-Aldrich, St. Louis, MO, USA), both from Sigma-Aldrich, as previously described (*Mueller, Millonig & Waite, 2009*). Briefly, the GOX and CAT enzymes were added to high glucose DMEM (4.5 g/L glucose; Thermo-Fisher, Waltham, MA, USA) and the resulting mixture placed on the confluent iMAEC for two hours of incubation. Hypoxia was subsequently terminated by removing the GOX/CAT DMEM and replacing it with normal DMEM containing 250 ng/ml of LPS. The iMAEC

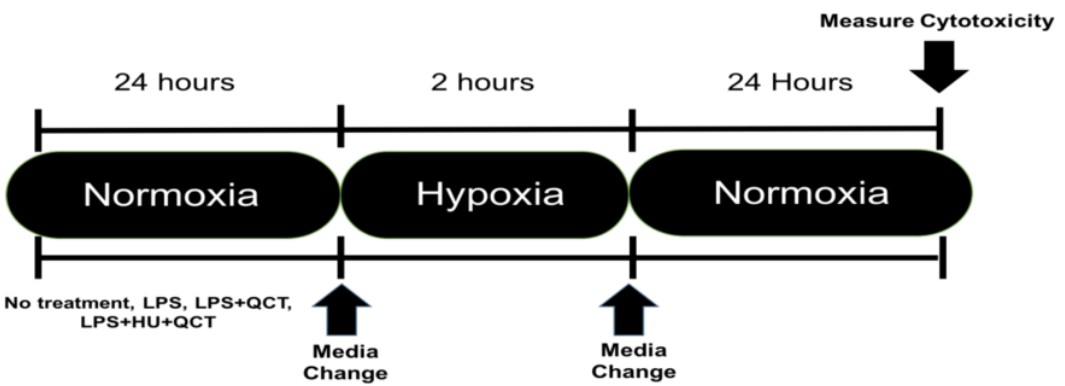

**Figure 1** **Experimental outline.** Timeline of experimental procedure.

were then allowed to incubate in the normal DMEM/LPS media for 24 h after which cytotoxicity was measured. (See Fig. 1 for outline of experiment)

## Measurement of cytotoxicity

LDH released from the cells into the medium, was measured as per the manufacturer's protocol (Pierce LDH Cytotoxicity Assay; ThermoFisher, Waltham, MA, USA), and served as a biomarker for cellular cytotoxicity following the cell treatments. Before starting the assay, 10 µl of 10X lysis buffer was added to healthy iMAEC cells and allowed to incubate at 37° for 45 min. These cells served as the positive control, indicating the expected mean concentration of LDH in each well. After 24 h of normal aerobic exposure, 50 µl of supernatant from each well was transferred to a new 96-well plate, 50 µl of substrate solution was added per the manufacturer's protocol and the cells were incubated at room temperature for 30 min. After 30 min, 50 µl of stop solution was added to each well. The absorbance for each well was then measured by spectrophotometer at 490 nm and 680 nm to account for background absorbance. To quantify LDH release, the absorbance of each treatment group was divided by the absorbance of the positive control group and multiplied by 100. Each result was expressed as a percentage of total LDH release.

## Statistical analysis

Results are presented as means $\pm$ SD. The D'Agostino's $K^2$ test was used to measure the distribution of the data. The data was not normally distributed; hence the Mann–Whitney $U$ Test was used to compare differences by treatment. Based on the results of the D'Agostino's $K^2$ test, the Kruskal–Wallis non-parametric analysis of variance with pair-wise comparison was utilized. Statistical significance was assumed at $p \leq 0.05$ for all analyses. No outliers, designated as SD $\pm$ 2, were identified. The data were analyzed using GraphPad Prism 7 for MacOS Sierra.

## RESULTS

To establish whether LPS (Inflammatory Model, 250 ng/ml) causes increased LDH release, two groups of cells were compared. The mean absorbance of the LPS treated cells was significantly higher than for the untreated cells ($0.60 \pm 0.17$ vs $0.42 \pm 0.12$, $p = 0.0004$)

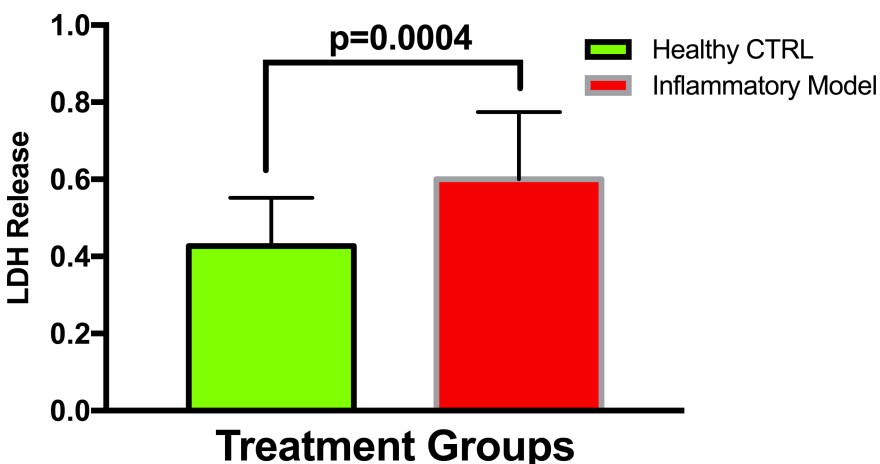

**Figure 2 Comparing LDH release in LPS treated cells versus non-treated cells.** Values are means ± SD. The mean LDH release of the LPS treated cells was significantly higher than for the untreated cells.

(Fig. 2). After confirming that LPS causes increased LDH release, we sought to determine whether various concentrations of QCT could prevent LDH release. The percentage of LDH released from the 30 μmol QCT-treated cells trended lower than the LDH% measured in the inflammatory model (LPS, 250 ng/ml) ($p = 0.1$, trend) (Fig. 3). Those cells treated with 100 μmol of HU, showed a significantly higher percentage of LDH release compared with LPS inflammation alone ($14.4 \pm 6.87$ vs. $5.67 \pm 5.73$, $p = 0.0004$). The addition of 30 μmol of QCT with the HU treatment gleaned a significant reduction of LDH release, compared with only HU treatment ($5.55 \pm 4.66$ vs. $14.4 \pm 6.87$, $p = 0.0002$). Interestingly, the combination treatment (QCT 30/100 μmol HU) returned LDH levels to those of the baseline LPS treated group (250 ng/ml), (Fig. 4).

## DISCUSSION AND CONCLUSIONS

Damage to vascular endothelial cells in SCD patients contributes considerably to the progression and severity of the disease (*Hebbel, Osarogiagbon & Kaul, 2004*; *Kaul, Finnegan & Barabino, 2009*). One form of injury to the endothelial cells results from cyclical periods of ischemia and reperfusion which exacerbate the endothelial cell damage and contribute to the progression of the disease (*Hebbel, 2014*). Since endothelial cells are a primary site of pathogenesis, there is considerable interest in developing treatments that target inflammatory mechanisms in these cells. This study was designed to determine if QCT, a plant flavonoid with anti-oxidant and anti-inflammatory properties, could reduce or prevent the damage caused by inflammation/hypoxia and damage caused by HU. Reports in the literature indicate that LDH is both a general marker of cellular damage and a marker of ischemia-reperfusion damage in SCD (*Stankovic Stojanovic & Lionnet, 2016*). This study used LDH as a marker for assessing the efficacy of QCT treatment for reducing inflammation in an in-vitro model. The first result confirmed that QCT (30 μmol) treatment showed a trend toward reducing the percentage of LDH released by an LPS induced inflammatory model undergoing a period of hypoxia and re-oxygenation. This

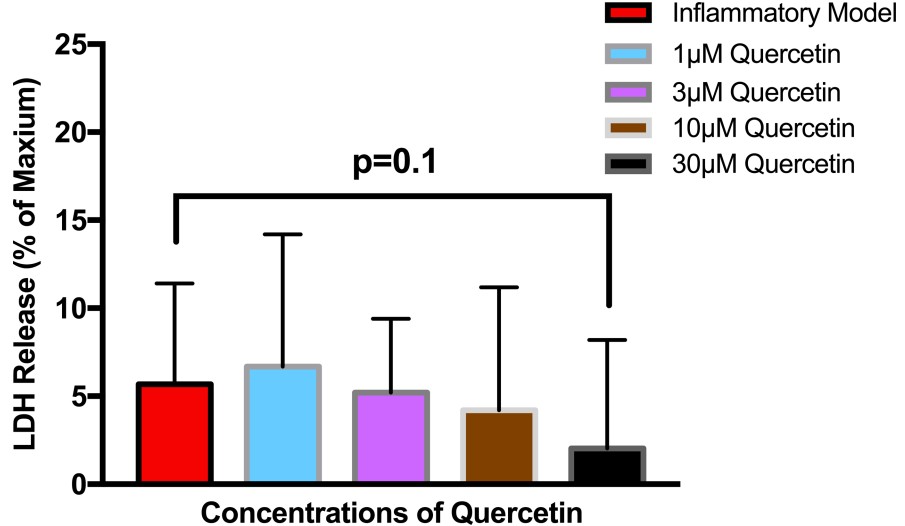

**Figure 3** **The effect of various concentrations of quercetin on LDH release.** Values are means ± SD. The percentage of LDH released from the 30 μmol QCT-treated cells trended lower than the LDH release measured in the inflammatory model.

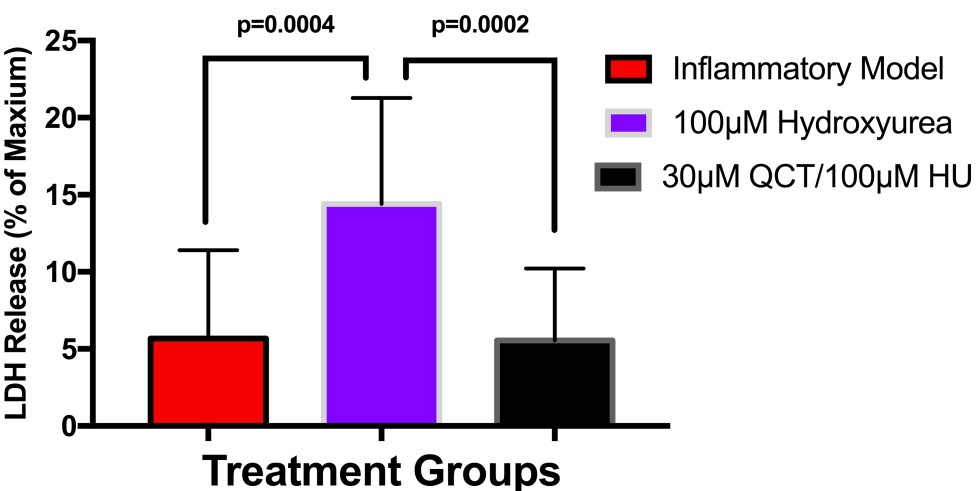

**Figure 4** **The effects of hydroxyurea and quercetin on LDH release.** Values are means ± SD. The addition of 30 μmol of QCT in combination with the HU treatment gleaned a significant reduction of LDH release, compared with only HU treatment.

lends preliminary support to the idea that QCT could be useful for reducing cytotoxicity caused by inflammation and oxidative stress in endothelial cells. The second finding that HU (100 μmol) significantly increased the amount of LDH released in the inflammatory model during a period of hypoxia and re-oxygenation is novel, since to our knowledge, there are no reports in the literature about this effect of low dose HU for treating SCD (*Brun et al., 2003*; *Baz et al., 2012*). By reducing the damage in endothelial cells, a key site of disease progression in SCD (*Osarogiagbon et al., 2000*; *Hebbel, Osarogiagbon & Kaul, 2004*; *Rajendran et al., 2013*; *Hebbel, 2014*), more severe disease side effects such as atherosclerosis

(*Elsharawy, Moghazy & Shawarby, 2009*), future clot formation, and vessel stenosis (*Adams, 2007*; *Doepp et al., 2012*; *Sparkenbaugh & Pawlinski, 2013*) could be alleviated or completely prevented. However, any reduction of baseline inflammation related to QCT treatment, could be reversed by subsequent inflammatory response.

While QCT may not completely avert damage caused by the combined effect of inflammation, oxidative stress and HU treatment, these findings suggest that the QCT may prevent injury from the HU drug, shown in these studies to cause inflammation, but which has previously been assumed only to reduce the subclinical ischemic events induced by sickle red blood cells. The combination of QCT and HU, resulted in significant reduction of LDH release to baseline levels of inflammation, indicating that QCT may completely prevent damage induced by HU treatment, possibly returning the endothelial cells to pre-treatment levels. Therefore, a combination treatment of QCT with the HU, may prevent or reduce HU treatment induced endothelial cell damage. Suggested QCT use as adjunct to HU therapy may translate to improved disease outcomes.

This study was limited to investigating only one type of endothelial cell, aortic endothelial cells. While there is vast homogeneity among endothelial cell types there are still subtle differences that may affect the outcome of QCT treatment (*Aird, 2012*). It is a vital future step to repeat these experiments in endothelial cells from several different locations in the body and of human origin. This could confirm QCT efficacy system wide. Once confirmed *in vitro* then in vivo models will need to be tested.

In summary, this study demonstrates a new and important finding that HU treatment could be increasing inflammation in the already chronically inflamed vascular endothelium in patients with SCD. The study successfully demonstrated that QCT may be an effective treatment in reducing cytotoxicity caused by inflammation resulting from ischemia-reperfusion, and by treatment with HU. Given that HU is an important pharmacological agent used in the treatment of SCD, it may advantageous for reducing possible short and long term cytotoxic effects associated with HU treatment. These studies show that pre-treatment with QCT may provide protection from endothelial cell damage caused by inflammation and hypoxic periods. Use of QCT as an adjunct treatment with HU could reduce the amount of HU required, limit endothelial cell injury and possibly decrease the severity of secondary disease consequences, such as stroke, vessel stenosis, and vasculopathy. Based on the results of this study, adding QCT to the standard treatment regimen should be seriously considered.

## ACKNOWLEDGEMENTS

We would like to give our gratitude to Dr. Sharon Francis for her guidance and support during these experiments. We also appreciate the members of Jo Lab at Emory University for their gift of the iMAEC and technical guidance.

### Funding

This work was supported by National Institutes of Health (RCMI G12 MBRC # 8G12MD0076), and National Institutes of Health (MBRS/RISE 2R25GM0582). This research is also supported by the RCMI G12 MBRC Program, Grant Number 8G12DM007602 from the National Institute of Minority Healthy and Health Disparities (NIHMD). The content is solely the responsibility of the authors and does not necessarily represent the official views of the NIMHD or the NIH. The funders had no role in study design, data collection and analysis, decision to publish, or preparation of the manuscript.

### Grant Disclosures

The following grant information was disclosed by the authors:
National Institutes of Health: RCMI G12 MBRC #8G12MD0076.
National Institutes of Health: MBRS/RISE 2R25GM0582.
MBRC Program: 8G12DM007602.
National Institute of Minority Healthy and Health Disparities (NIHMD).

### Competing Interests

The authors declare there are no competing interests.

### Author Contributions

- Zachary M. Kiser conceived and designed the experiments, performed the experiments, analyzed the data, contributed reagents/materials/analysis tools, wrote the paper, prepared figures and/or tables, reviewed drafts of the paper.
- Monica D.M. McGee conceived and designed the experiments, performed the experiments, contributed reagents/materials/analysis tools.
- Racquel J. Wright conceived and designed the experiments, performed the experiments.
- Alexander Quarshie analyzed the data, contributed reagents/materials/analysis tools.
- Gale W. Newman, Karen R. Randall, Jonathan K. Stiles and Adel Driss reviewed drafts of the paper.
- Jacqueline M. Hibbert conceived and designed the experiments, analyzed the data, contributed reagents/materials/analysis tools, wrote the paper, reviewed drafts of the paper.

### Data Availability

Raw data can be found in the Supplemental Information.

### Supplemental Information

Supplemental information for this article can be found online at http://dx.doi.org/10.7717/peerj.3376#supplemental-information.

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
