# Peer review of "Quercetin reduces hydroxyurea induced cytotoxicity in immortalized mouse aortic endothelial cells"

_PeerJ, doi:10.7717/peerj.3376_

## Round 0.1 · original submission · Minor Revisions

I believe that addressing the reviewer comments will improve the quality of the paper.

Reviewer 1 ·

Basic reporting

 Literature references, sufficient field background/context provided:
I added a reference in the Introduction . It is marked in red in the corrected attached file
The reference is : L.Potenza, C.Calcabrini, R. De Bellis, U. Mancini, L. Cucchiarini, M.Dachà .Effect of quercetin on oxidative nuclear and mitochondrial DNA damage.BioFactors , 33: 33–4, 2008.
 Professional article structure, figs, tables:
I’d preferred better legends to figures . They were copied from the section “Results”. For example authors can name figure 2: LDH release. Then they comment results. In the y axis of this figure I’d replace Absorbance with ”LDH release”

Experimental design

The Experimental design is correct

Validity of the findings

 Data is robust, statistically sound, & controlled:
the authors write in the Discussion and Conclusions: “The second finding that HU (100μmol), significantly increased the amount of LDH released in the inflammatory model, during a period of hypoxia and re-oxygenation, is novel since to our knowledge, there are no reports in the literature about this effect of low dose HU for treating SCD” Are the authors sure when they write “low dose HU (100µmol)? They work on 20.000 cells .Is really a low dose? In literature I found only reports on human were HU is introduced as mg/kg/day. When a drug is administered it is already studied on cellular and animal models. A cytotoxicity test is the first step to study a drug : it seems strange that this effect wasn’t already observed. May it depend on cell line selected?
 Speculation is welcome, but should be identified as such
In my opinion there were too many speculations

Additional comments

The manuscript reports the protective effect of quercetin (QCT) on Hydroxyurea (HU) induced cytotoxicity in immortalized mouse aortic endothelium cells. In this study the authors performed an in vitro model of inflamed vascular endothelium using Lipopolysaccarides (LPS). Combined treatment were made with HU, QCT or QCT+ HU and LDH level was analysed as marker of cellular damage. The authors found that HU (100 µM) significantly increased the amount of LDH release in the set up model and combination of QCT and HU led to baseline levels of inflammation indicating that QCT may completely prevent damage induced by HU treatment.
The manuscript is well written, the experimental design is correct and results are clearly presented. This work is quite limited analysing only a damage marker but represents a starting point which can led to use quercetin as an adjunct treatment. Further investigations should be performed on the set up model to better evaluate the protective effect of QCT : ROS level, Inflammatory cytochines and adhesion molecules, SOD, GSH……..
Animal models will be requested in order to understand QCT bioavailability. However I think that it is fair to inform the scientific community about these two obtained results: low dose of HU increase cytotoxicity and QCT reverse to basal level of inflammation. From these knowledge further study must be designed to better understand HU and QCT molecular mechanisms and finally reduce HU side effects. For these my reasons I think that manuscript may be accepted following my suggestions also marked in red in the attached file

Annotated reviews are not available for download in order to protect the identity of reviewers who chose to remain anonymous.

·

Basic reporting

- Some literature references are very old. Since SCD is a well-studied subject, there are several current revisions that can be used.
- The legends of the figures are repeating the experimental design of the study. I suggest that all the information regarding the experimental phase be placed in the material and methods section.
- Figure 4 is citing "the most effective dose from Figure 2", but I think it´s dose from Figure 3, isn´t it?

Experimental design

No comment.

Validity of the findings

No comment.

Additional comments

As SCD is a serious public health problem in many countries, any data that can improve the quality of life of the patient being treated is of paramount importance.

---

## Round 0.2 · accepted · Accept

All the revisions requested have been correctly performed.